# Protocol for a multicentre, parallel-group, open-label randomised controlled trial comparing ferric carboxymaltose with the standard of care in anaemic Malawian pregnant women: the REVAMP trial

Martin N Mwangi [1,2] Glory Mzembe,[1,2] Ernest Moya,[1,2] Sabine Braat [3,4] Rebecca Harding,[5] Bjarne Robberstad,[6] Julie Simpson,[3] William Stones [7] Stephen Rogerson,[4,8] Kabeya Biselele,[9] Jobiba Chinkhumba [10] Leila Larson,[11] Ricardo Ataíde,[4,5] Kamija S Phiri,[1,2] Sant-Rayn Pasricha[5,12]

KSP and S-RP are joint senior authors.

For numbered affiliations see end of article.

**Correspondence to**
Dr Martin N Mwangi;
mmwangi@true.mw

## ABSTRACT

**Introduction** Anaemia in pregnancy remains a critical global health problem, affecting 46% of pregnant women in Africa and 49% in Asia. Oral iron therapy requires extended adherence to achieve correction of anaemia and replenishment of iron stores. Ferric carboxymaltose (FCM) is a recently established intravenous iron formulation associated with substantial advantages in safety, speed of delivery and total dose deliverable in a single infusion. We aim to determine whether FCM given once during the second trimester of pregnancy compared with standard oral iron distributed through routine antenatal services is effective and safe for treatment of moderate to severe maternal anaemia in sub-Saharan Africa.

**Methods and analysis** The randomized controlled trial of the effect of intravenous iron on anaemia in Malawian pregnant women (REVAMP) is a two-arm confirmatory individually randomised trial set in Blantyre and Zomba districts in Malawi. The trial will randomise 862 women in the second trimester of pregnancy with a capillary haemoglobin concentration below 100.0 g/L. The study comprises two arms: (a) intravenous FCM (20 mg/kg up to 1000 mg) given once at randomisation, and (b) standard of care oral iron (65 mg elemental iron two times per day) for 90 days (or the duration of pregnancy, whichever is shorter) provided according to local healthcare practices. Both arms receive sulfadoxine-pyrimethamine as intermittent preventive treatment in pregnancy. The primary outcome is the prevalence of anaemia (Hb <110.0 g/L) at 36 weeks' gestation. Secondary outcomes include birth weight, gestation duration and safety outcomes, including clinical malaria, serious perinatal events and postpartum haematologic and health-related outcomes in the mother and child.

**Ethics and dissemination** Ethical approval was granted by the Research Ethics Committee (COMREC P.02/18/2357) in Malawi and the Human Research Ethics Committee (WEHI: 18/02), Melbourne, Australia. The protocol is registered with the Australian and New Zealand

## Strengths and limitations of this study

► Eligibility for inclusion—moderate or severe anaemia—is assessed by capillary haemoglobin estimation, a method that could be deployed at the local health centre level.

► The trial follows pregnant women and their babies through delivery and into the postpartum period enabling the assessment of antenatal and postnatal effects of the intervention.

► This trial uses a modern intravenous iron formulation for the treatment of anaemia in pregnancy which enables a high dose iron infusion (up to 1000 mg) to be infused in a single dose.

► The trial will measure a broad range of haematologic, safety and clinical-efficacy outcomes.

► This trial is open-label, and participants will know the trial intervention to which they have been randomised.

Clinical Trials Registry. The results will be shared with the local community that enabled the research, and also to the international fora.

**Trial registration number** ACTRN12618001268235; Pre-results.

## INTRODUCTION

Anaemia during pregnancy remains a critical global health problem. Almost 40% of pregnant women worldwide are anaemic, including 46% of pregnant women in Africa and 49% in Asia.[1] Anaemia in pregnancy is very common and mostly results from iron deficiency and is associated with critical risks for both mother (eg, life-threatening complications of postpartum haemorrhage)

and child (especially prematurity and low birth weight,[2] which are associated with increased risk of neonatal and infant mortality,[3] and reduced iron stores in infancy with increased risk of subsequent anaemia and impaired development).[4] Severe anaemia in pregnancy is associated with a significantly increased risk of maternal mortality.[5] Worldwide, 15%–20% of births (>20 million annually, including 13% in sub-Saharan Africa) are low birth weight. At the same time, each year, over 1 million children die from preterm birth complications, making prevention and treatment of antenatal anaemia a crucial component of efforts to reduce low birth weight.

Systematic reviews confirm likely benefits from iron on maternal outcomes, including anaemia (70% reduction) and trends towards favourable infant outcomes, including increased birth weight and extended gestation duration.[6] A trial of oral iron compared with placebo in Kenyan pregnant women confirmed a benefit from iron on birth weight and gestation duration.[7] Global recommendations for managing anaemia in pregnancy in lower and middle income countries currently advise that women be treated with high-dose daily oral iron (120 mg of elemental iron) supplementation for 3 months.[8 9] However, oral iron is often poorly tolerated due to gastrointestinal adverse effects,[10] limiting adherence; lower doses with intermittent dosing have been used in non-pregnant women.[11] Delivery of iron during pregnancy requires high-quality performance of the primary health system. In Malawi, 10.6% of women were reported to take no iron during pregnancy, and 37.1% take less than 60 doses, with only 33.4% taking the recommended 90 or more doses.[12] Even when delivered, oral iron frequently fails to correct anaemia. For example, 61% of Gambian pregnant women who received iron at week 20 of gestation—when their Hb was lower than 100.0 g/L—still had a Hb <100.0 g/L at 30 weeks' gestation.[13] Finally, women may only present for their first antenatal visit late in the second trimester, curtailing the time available to optimise iron stores with oral iron to improve foetal development and risks associated with delivery; for example, in Malawi, fewer than 30% of pregnant women attend before the 16th week of gestation.[14]

Over the past decade, there have been dramatic improvements in parenteral (intravenous) iron therapeutics. Ferric carboxymaltose (FCM) is a widely used formulation that can be administered in a short period (15 min) and at large doses (up to 1000 mg) in a single infusion.[15 16] FCM has revolutionised the treatment of iron-deficiency anaemia in high-income country settings and is widely used in outpatient settings, emergency departments and primary care.[17] In pregnancy, intravenous iron has been reported to be superior to oral iron in improving haemoglobin concentrations and may even increase birth weight.[18] Intravenous iron is increasingly recommended as a suitable first-line option for women with moderate or severe iron-deficiency anaemia beyond the first trimester of pregnancy.[19] Studies have evaluated the role of older forms of intravenous iron in pregnancy in low-income settings such as India,[20 21] while other studies have demonstrated the feasibility of using FCM in the postpartum period in sub-Saharan Africa[22] but modern formulations capable of delivering a rapid total-dose infusion have not yet been studied in women in pregnancy in low-income countries.

It remains challenging to directly measure iron status in the field, as biomarkers generally require analysis using centralised laboratory infrastructure. In contrast, point-of-care measurement of capillary haemoglobin is a feasible field-friendly testing strategy that can be deployed with existing technology, of which HemoCue point-of-care devices are a good example. Screen-and-treat approaches for anaemia in pregnancy must rely on haemoglobin measurement until point-of-care assays for iron status become more widely available.

FCM presents a practical opportunity for rapid correction of moderate and severe antenatal anaemia in a single dose, potentially improving maternal and neonatal clinical outcomes for pregnant women. It is, therefore, urgent to test this therapy in a low-income field setting where the burden of anaemia-related disease is heaviest. In Malawian pregnant women with moderate or severe anaemia during the second trimester of pregnancy, we aim to evaluate whether a single dose of intravenous FCM is superior to oral iron provided via standard-of-care approaches.

## METHODS AND ANALYSIS
### Patient and public involvement
We held discussions with national policy stakeholders during the study planning stage, including the Ministry of Health. We discussed how the research might align with national research and health service priorities. Local community engagement was done via public meetings; the potential participants were first involved in the study's design during these meetings. We discussed the study with the traditional leaders, including traditional authorities, group village headmen (GVH), the village development committee and ward councillors (political figures). The GVH were requested to cascade the information to the village chiefs who took the information to the community. Health workers were accessed via the District Health Offices in Blantyre and Zomba districts. Health workers from the participating health facilities, including the health surveillance assistants who directly work with the community, were informed about the proposed research and discussions on priorities beneficial to the community were discussed. The public, potential participants and health workers identified malaria and anaemia as major public health issues in the community. Women outlined their experiences with iron supplements during their past pregnancies and identified the development of tolerable iron formulations as a research priority. Dissemination of findings at the national and community levels will follow

the schema used during study design and inception as outlined above.

## Trial objectives

The primary objective of the trial is to determine whether, in Malawian women in the second trimester of pregnancy with moderate or severe anaemia, a single dose of FCM up to 1000 mg is superior to standard-of-care (ie, oral iron provided through local health services), in reducing maternal anaemia before delivery (at 36 weeks' gestation).

The secondary objectives are:

During pregnancy and up to 1 month postpartum, to determine the effects of FCM (compared with standard-of-care) on:

*Effectiveness*
1. Maternal haemoglobin concentration and iron status (measured through iron biomarkers).
2. Critical neonatal outcomes including birth weight (low birth weight), gestation duration (prematurity), small for gestational age and other perinatal outcomes.

*Safety*
1. Maternal and neonatal adverse events (AEs), including infection episodes, serious maternal complications and hypophosphatemia.

Exploratory objectives up to 1 month postpartum for subsequent hypothesis generation are related to maternal cognition, depression and fatigue, as well as costs of healthcare.

## Study design

The Randomized controlled trial of the Effect of intraVenous iron on Anaemia in Malawian Pregnant women (REVAMP) is a multicentre, open-label, two-arm, parallel-group individually randomised controlled trial (RCT). An open-label (rather than blinded) design was selected as it was considered unfeasible to deliver the placebo intravenous infusions to Malawian pregnant women in this field-trial context (the drug is dark-coloured). The trial recruited pregnant women in their second trimester and is following them until 1 month postpartum, after which we will report on the primary and secondary objectives. Extended follow-up of mothers and infants to 12 months postpartum is planned, as is a range of exploratory economic, biological and clinical outcomes. These analyses are beyond the scope of this protocol, as are the exploratory outcomes collected up to 1 month postpartum, and they will be reported separately. The trial design is summarised in figure 1.

## Study settings and participants

In Malawi, the average gestation age at the first antenatal clinic (ANC) visit is 22 weeks, with 61% presenting before 24 weeks. The 2015–2016 DHS survey estimated a prevalence of anaemia (Hb <110.0 g/L) in pregnant women in Malawi at 45.1%, including 22.4% of women with moderate or severe anaemia.[12] The prevalence of low birth weight in Malawi was around 14% in 2015.[23] Malawi is endemic for malaria and the prevalence of *Plasmodium* spp parasitaemia at the first ANC visit in the study site exceeds 15%.[24] About 87% of women consume at least one dose of intermittent preventive treatment during pregnancy (IPTp), but this falls to 30% for women taking three or more doses.[12]

The REVAMP study has two major research sites namely Limbe Health Centre and Zomba Central Hospital located in Blantyre and Zomba districts,

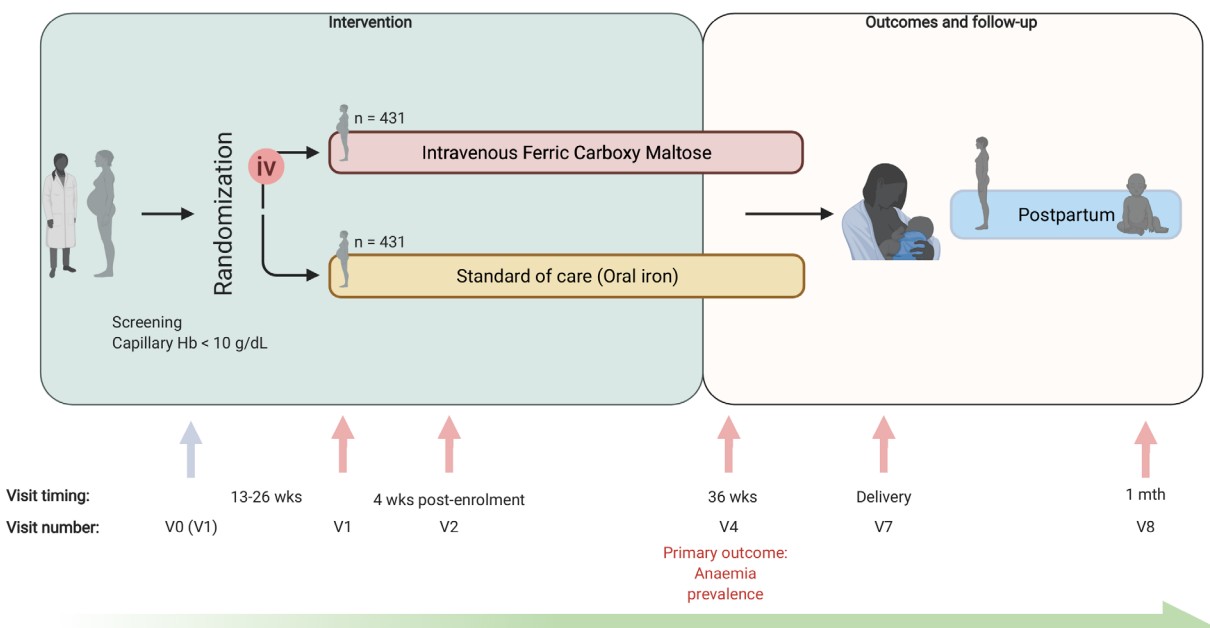

**Figure 1** REVAMP trial schema. iv, intravenous; REVAMP, Randomized controlled trial of the Effect of intraVenous iron on Anaemia in Malawian Pregnant women; V0, visit 0; V1, visit 1; V2, visit 2; V4, visit 4; V7, visit 7; V8, visit 8; wks, weeks.

respectively, in Southern Malawi. The two major sites act as the study coordination centres in the respective districts. The Blantyre site serves as the base for five urban health centres/ANCs, while the Zomba site serves as the base for nine (four urban and five rural) health centres/ANCs. Women are screened at the ANCs and referred to the coordinating research site for enrolment, treatment and follow-up visits. Deliveries occur in government-operated birth suites attached to the study site with referral to a district hospital where required for obstetric indications.

### Eligibility criteria
Women are only randomised if they fulfil all inclusion criteria and none of the exclusion criteria.

### Inclusion criteria
1. Confirmed singleton pregnancy at 13–26 weeks' gestation.
2. Moderate to severe anaemia (capillary haemoglobin <100.0 g/L measured by HemoCue 301+) not clinically deemed to require an immediate blood transfusion. We excluded women with mild anaemia as thresholds to distinguish mild anaemia from health are indistinct, and because moderate and severe anaemia have an increased link to adverse maternal and child health outcomes.[25]
3. Negative malaria parasitaemia at screening and no positive malaria rapid diagnostic test within the previous 7 days.
4. Resident in the study catchment area of Blantyre and Zomba district.
5. Plan to deliver at a health facility.
6. Written informed consent, including assent if <18 years old (consent of a parent or guardian plus the participant's assent).

### Exclusion criteria
1. Hypersensitivity to any of the study drugs.
2. Clinical symptoms of malaria or positive malaria rapid diagnostic test within the previous 7 days, or symptoms of bacterial infection, at screening.
3. Any condition requiring hospitalisation or serious concomitant illness.
4. Chronic illness that may adversely affect foetal growth and viability.
5. Severe anaemia clinically judged as requiring a blood transfusion (usually Hb <50.0 g/L).
6. Pre-eclampsia at screening (new onset of hypertension, proteinuria and swelling in the legs, feet and hands).

### Trial interventions
Participants are randomly assigned to receive one of the following interventions:
1. *Intravenous iron*: FCM 20 mg/kg up to 1000 mg, diluted in normal saline and given over 15 min, once after randomisation. The study clinician administers the drug, and women remain under observation for 45 min after drug administration. Women receiving FCM are treated in rooms equipped with a trolley containing epinephrine, hydrocortisone, intravenous fluids and antihistamines, airway equipment and an oxygen concentrator. Staff are trained in the management of allergic reactions. FCM, manufactured by Vifor Pharma, was purchased at full-price from Aspen Pharma in Australia and shipped to Malawi.
2. *Standard-of-care—oral iron treatment course*: oral iron-200 mg ferrous sulphate (approx. 65 mg elemental iron) two times per day for 90 days or the remaining duration of pregnancy, whichever is shorter. Oral iron is provided by health workers using the same education messaging and similar packaging as provided by the routine health system to reflect real-life health service conditions.

Following national guidelines, both groups receive IPTp with 1500 mg sulfadoxine and 75 mg pyrimethamine (SP), as three tablets of SP of fixed dose (500 mg/25 mg) at least three times during pregnancy, at least 4 weeks apart. The IPTp-SP course at 4 weeks' post-randomisation is directly observed by study staff. HIV positive women in Malawi are not offered SP because they are already on cotrimoxazole prophylaxis. The baseline dose is also directly administered by study staff if the participant did not receive it through the health service. All participants also receive an insecticide-treated bed net.

### Randomisation, allocation concealment and blinding
Participants are randomly allocated to one of the two treatments arms with 1:1 allocation via a computer-generated randomisation schedule of randomly permuted blocks stratified by research site (Blantyre, Zomba) to achieve a balance between the arms within each site. The randomisation list was generated by an independent statistician at the University of Melbourne (Australia).

Individual participant codes were pre-packed in sealed envelopes by an independent researcher not associated with the study and held securely at the research sites. Eligible participants who meet all inclusion/exclusion criteria are sequentially allocated participant identification numbers within the research site, and their allocation to study group is revealed after opening the corresponding sealed envelope. Although the trial is open-label, laboratory scientists measuring haemoglobin concentration, midwives collecting birth outcome data, and investigators and researchers in Australia (including data managers and statisticians in Melbourne) are blinded to the treatment allocation during the conduct of the trial until the database is locked and ready for unblinding.

### Recruitment and visits
Table 1 shows the schedule of activities per visit for the mother, and table 2 shows the schedule of activities per visit for the neonate.

### Screening and enrolment of participants
Study recruitment opened on 12 November 2018; the final participant was recruited on 2 March 2021. Study activities are detailed in tables 1 and 2.

**Table 1** Planned activities per visit for mothers

| Protocol activity | Visit 0 | Visit 1 | Visit 2 | Visit 3* | Visit 4 | Visit 5–6* | Visit 7 | Visit 8 |
|---|---|---|---|---|---|---|---|---|
| | Day −7 to 0 | Day 0 | Day 28±2 days | 34 wks gestation ±2 days | 36 wks. gestation ±2 days | 38–40 wks gestation ±2days | Delivery +1 day | Day 28 postpartum ±2days |
| Location of visit | Antenatal clinic | Research site† | Research site† | Home‡ | Research site† | Home‡ | Research site† | Research site† |
| Informed consent process for pre-screening | X | | | | | | | |
| Pre-screening | X | | | | | | | |
| Detailed informed consent process | X | X | | | | | | |
| Screening | | X | | | | | | |
| Medical and obstetric history | X | X | | | | | | |
| Demographics | | X | | | | | | |
| Maternal physical examination§ | | X¶ | X** | | X** | | X§ | X** |
| Ultrasound scan (foetal biometry)§ | | X | X | | X | | | |
| Randomise participant | | X | | | | | | |
| Administer treatment | | X | | | | | | |
| Intermittent preventive treatment†† | | X | X | | X | | | |
| Laboratory procedures | | | | | | | | |
| Capillary haemoglobin§ | | X | X | X | X | X | X | X |
| Full blood count§ | | X | X | | X | | X | X |
| Malaria diagnostics§‡‡ | X | X | X | | X | | X | X |
| Serum for iron markers tests§§ | | X | X | | X | | X | X |
| Serum for inflammatory markers tests¶¶ | | X | X | | X | | X | X |
| Phosphate | | X | X | | X | | X | X |
| Adverse events§ | | X | X | | X | | X | X |
| Morbidities | | | | | | | | X |
| Missed visits§ | | | X | X | X | X | X | X |
| End of study | | | X | X | X | X | X | X |

*Since the opening of the trial, visits 3, 5 and 6 have been suspended since April 2020 due to the risk of exposing communities to COVID-19 during home visits.
†Visit conducted at the research site/health facility.
‡Visit conducted at the participant's home.
§Protocol activities are also collected at any unscheduled visits.
¶Complete maternal physical examination includes general appearance, throat, neck, thyroid, musculoskeletal, skin, lymph nodes, extremities, pulses, pulmonary, cardiac, abdominal and neurological examination.
**Limited maternal physical examination includes general appearance, brief pulmonary, cardiac, abdominal and neurological examination.
††Assessing if intermittent preventive treatments sulphadoxine pyrimethamine and albendazole were given.
‡‡Malaria diagnostics include malaria rapid diagnostic test, malaria microscopy.
§§Serum for iron markers tests include serum ferritin.
¶¶Serum for inflammatory markers tests include C reactive protein and alpha-1 glycoprotein.

**Table 2** Details of planned activities per visit for neonates

| Protocol activity | Visit 7 Delivery +1 day | Visit 8 Day 28 postpartum±2 days |
|---|---|---|
| Location of visit | Research site* | Research site* |
| Pregnancy outcome | X | |
| Physical examination and anthropometry†‡ | X§ | X¶ |
| Laboratory procedures | | |
| Capillary haemoglobin | X** | |
| Full blood count† | X** | X |
| Malaria diagnostics†† | X** | X |
| Serum for iron markers tests‡‡ | X** | X |
| Vaccination and Vit A supplementation status | | X |
| Adverse events† | X | X |
| Morbidities | | X |
| Missed visits† | | X |
| End of study | X | X |

*Visit done at the research site/ health facility.
†Protocol activities are also collected at any unscheduled visits.
‡Anthropometry includes assessing the infant's weight, length and head circumference.
§The complete physical examination includes general appearance, throat, neck, thyroid, musculoskeletal, skin, lymph nodes, extremities, pulses, pulmonary, cardiac, abdominal and neurological examination.
¶The limited examination includes general appearance, brief pulmonary, cardiac, abdominal and neurological examination.
**These assessments are collected via cord blood.
††Malaria diagnostics include malaria rapid diagnostic test, malaria microscopy, malaria filter paper for PCR and histology at delivery.
‡‡Serum for iron markers tests include serum ferritin.

### Pre-screening and screening, visit 0 (day −7 to 0)

Women attending their first antenatal visit at ANCs are given general information (aims and procedures) about the study by a trained research nurse and invited to consent to pre-screening procedures. As the ANC nurse cares for attendees, the research nurse pre-screens potential participants by noting apparent gestation age based on clinical examination and last menstrual period and then tests for anaemia with capillary Hb (HemoCue 301+, Angelholm Sweden) and *Plasmodium* infection by rapid diagnostic test (SD Bioline Malaria AG P.F/PAN, Standard Diagnostics). Potentially eligible participants receive further information about the study. Willing participants are referred to the research site for additional evaluation, full consenting procedure and recruitment.

### Enrolment, visit 1 (day 0)

As presented in table 1, at the health facility, potentially eligible participants undergo ultrasound scanning (within 2 days from the initial pre-screening visit) to confirm singleton pregnancy and determine gestation age (ie, crown-rump length for less than or equal to 16 weeks, femur length and head and abdominal circumference for more than 16 weeks). Written informed consent is then obtained for participation in the trial. For consenting participants, a physical examination is conducted, and baseline data are collected, comprising demographic information and medical and obstetric history. Following the collection of baseline clinical data, samples of both capillary and venous blood are collected and then participants are randomised. For women randomised to the FCM arm of the trial, an intravenous cannula is inserted under aseptic precautions, and a trained nurse administers the study drug according to standard procedures and under the supervision of the study clinician. Women randomised to the standard-of-care arm of the trial are provided with a full course of iron tablets, together with information delivered according to a standardised script reflecting ANC practice.

### Subsequent study visits

#### Visit 2 (28 (±2) days after enrolment into the study)

Participants receive a physical examination and an ultrasound scan before being asked for a venous blood sample (table 1). A capillary sample is collected for assessment of *Plasmodium* parasitaemia. The participants are asked to continue their regular ongoing ANC visits through their local health centre.

#### Visit 3; visits 5–6 (34 weeks' gestation ±2 days; 38–40 weeks' gestation ±2 days)

Participants are visited in their home at 34 weeks' gestation, and every 2 weeks from 38 weeks' gestation until delivery. A research nurse measures capillary Hb. Where

the participant belongs to the standard-of-care trial-arm, the research nurse attempts to assess adherence. Notably, since the opening of the trial, fieldwork has been affected by the COVID-19 pandemic. Because of the risks of study workers introducing COVID-19 into remote villages, home visits were removed from the trial protocol after April 2020.

### Visit 4 (36 weeks' gestation/pre-delivery±2 days)

Procedures are similar to those for visit 2 (28-days post-treatment). This visit includes the assessment of venous haemoglobin for the primary outcome.

### Visit 7 (delivery +1 day)

The study provides 24-hour cover of the study research sites' delivery suites. All participants are asked to return to the research site for delivery (unless a high-risk pregnancy requires delivery at the tertiary referral hospital). Standardised delayed cord clamping procedures are instituted. Participants delivering at home or at other health facilities are encouraged to attend the research site for assessment within 24 hours. Venous blood samples are collected during labour. Apgar scores are recorded immediately after delivery, and the newborn undergoes a full physical examination, including measurement of birth weight, length, head circumference and assessment for congenital malformations. The type of birth and occurrence of perinatal complications (including haemorrhage or need for blood transfusion) are recorded. Placental blood is drawn, and a sample of placental tissue is collected and stored in buffered formalin for subsequent histological evaluation.

### Visit 8 (28 days postpartum ±2 days)

Participants return to the research site together with their infants for a detailed medical examination of both mother and infant and collection of blood samples.

### Unscheduled visit

Participants are asked to attend the research site when symptomatically unwell. They are managed according to national standard treatment guidelines by a trained healthcare provider. Blood samples for malaria Rapid Diagnostic Test (RDT) (and microscopy if RDT positive), and blood culture are taken if clinically indicated. The clinical diagnosis of the unscheduled visit is recorded in the participant's record. Where a participant attends another health facility or ANC, the research team extracts the missed unscheduled visit notes from the participants' health book commonly known as a health 'passport' in Malawi. Usually, clinical investigations and medications are indicated in the health book and can be extracted by the research team during the next scheduled visit.

### Laboratory procedures

Venous blood is measured for haemoglobin concentration using an automated analyser (Sysmex, XP 300 series, Sysmex Corporation, Kobe, Japan), for which daily two level controls are run and recorded. Serum is separated

by centrifugation and stored at −80°C. Samples will be batched and assayed for ferritin, C reactive protein and phosphate in Meander Medical Centre laboratory, accreditation number M040, EN ISO 15189:2012 (Amersfoort, The Netherlands).

### Data monitoring committee (DMC)

An independent DMC has been set up to review on a regular basis, safety and efficacy data of the ongoing trial. The DMC is comprised of international experts in clinical trials, obstetrics, epidemiology and statistics. In the advent of clear evidence of effectiveness or harm, the DMC will recommend to the sponsor and investigators whether to continue, modify or terminate the trial on ethical grounds.

### Outcomes

The primary outcome is the prevalence of maternal anaemia before delivery, defined as a venous haemoglobin concentration less than 110.0 g/L at 36 weeks' gestation. This outcome evaluates the performance of the study intervention in helping women reach labour with optimal tissue oxygenation and resilience.

Secondary outcomes assessing effectiveness in the mother are anaemia at 4 weeks' post-randomisation, delivery, and 1 month postpartum, moderate/ severe anaemia and haemoglobin concentration both at 4 weeks post-randomisation, 36 weeks' gestation, delivery and 1 month postpartum, iron deficiency (ferritin <15 mg/L adjusted for C reactive protein), iron-deficiency anaemia and serum ferritin concentration all at 4 weeks post-randomisation, 36 weeks' gestation, and 1 month postpartum.

Effectiveness in the neonate is assessed using the outcome of mean birth weight (measured in grams within 24 hours of birth) and the secondary outcomes low birth weight (birth weight less than 2500 g), gestation duration (weeks), birth length, small for gestation age, fetal loss (pregnancy loss before 28 completed weeks' gestation), stillbirth (birth of a child showing no signs of life after 28 weeks' gestation), preterm births (neonate born before 37 completed weeks of gestation), haemoglobin concentration at 1 month postpartum and infant growth (weight for age, length for age, weight for length) at 1 month postpartum.

Outcomes assessing safety in the mother are (serious) AEs related to administration of FCM (events occurring during the enrolment visit), (serious) AEs (within 14 days of randomisation, antenatal and postpartum reported separately), all-cause sick visits to the clinic (antenatal and postpartum reported separately), clinical malaria-specific visits to the clinic (antenatal and postpartum reported separately), hypophosphatemia (biochemical) (at 4 weeks' post-randomisation and 36 weeks' gestation), inflammation (elevated C reactive protein) (at 4 weeks' post-randomisation and 36 weeks' gestation) and severe medical events including haemorrhage, receipt of blood

transfusion, Intensive Care Unit (ICU) admission or mortality.

Clinical infections will be reported during unplanned visits. Clinical malaria will be defined clinically, in women who present with fever and a positive malaria test. Diarrhoea will be defined in women with more than three loose stools per day. Other clinical diagnoses will be made according to local health manuals.

Safety in the infant is assessed using (serious) AEs, all-cause visits to the clinic, infection-related visits to the clinic, diarrhoea-related visits to the clinic and clinical malaria-specific visits to the clinic.

In addition to the primary and secondary clinical outcomes listed above, which will support the reporting of the confirmatory RCT, data for many exploratory outcomes are being collected between enrolment and 12 months postpartum. These include assessing maternal cognition, mother–infant bonding, maternal depression and fatigue, infant neurodevelopment, infant diet and health economic data, including direct and indirect costs of healthcare. Also, women have given their consent for the collection of samples for future translational studies, including evaluation of the vaginal and faecal microbiome, maternal immune profiles and effects of interventions directed towards *Plasmodium falciparum* biology. Details of primary, secondary and exploratory outcomes included in the trial registration are shown in online supplemental table 1.

### Detection and reporting of AEs and serious AEs (SAEs)

Non-serious AEs and SAEs are collected from the time consent is given until the participant completes the study (the final visit or withdrawal). These are detected either through spontaneous reports by the participant, unplanned visits to the research site or any of the participating health centres, observation by the study staff, and through standard questioning at each visit. All SAEs are reported to the sponsor and the appropriate ethics committee, whether or not considered causally related to the study drugs.

### Sample size

The sample size calculation is based on the primary maternal outcome of the proportion of women with pre-delivery anaemia (defined as venous Hb <110.0 g/L at 36 weeks' gestation). A systematic review and meta-analysis identified a 50% reduction in anaemia prevalence in women receiving oral iron.[2] In a cohort of pregnant women in the Gambia, we observed that 60% of pregnant women with Hb <100.0 g/L at 20 weeks' gestation who then receive iron interventions continue to have Hb <100.0 g/L by 30 weeks.[13] The pivotal FER-ASAP (FERric carboxymaltose – Assessment of SAfety and efficacy in Pregnancy) trial (which compared FCM to oral iron in women with iron-deficiency anaemia in high-income settings) demonstrated a 14% reduction in absolute anaemia prevalence compared with oral iron.[26] We hypothesise that routine iron supplementation will reduce anaemia prevalence from 100% to 60% (as seen in the Gambia,[13] and similar to data from Haider and colleagues[2]). We hypothesise that FCM will result in a 10% absolute improvement in anaemia cure (to a prevalence of 50%). A total sample size of 862 pregnant women or 431 per group will provide 80% power to detect this difference, allowing for a 10% loss to follow-up and assuming a two-sided alpha of 5%. The sample size also has at least 80% power to detect an absolute difference between standard-of-care oral iron and FCM of 100 g in the neonatal outcome of birth weight (assuming an SD of 450 g and a two-sided alpha of 5%), similar to the effect size seen in a trial of Kenyan women receiving oral iron when compared with placebo.[7]

## STATISTICAL ANALYSIS PLAN

A detailed statistical analysis plan will be finalised before unblinding of the database. Analyses will be performed where participants are classified according to their randomised intervention arm (ie, intention to treat principle). An available case analysis will be performed for repeated time point outcomes (eg, anaemia) and a complete-case analysis for single time point outcomes (eg, birth weight). Anaemia will be analysed using a log-binomial regression model, including study participants as a random intercept to account for the multiple time points. The model will include the standard-of-care (oral iron) group as the reference group. The primary maternal hypothesis will be evaluated by obtaining the estimate of the prevalence ratio of intravenous iron versus standard-of-care (oral iron), 95% CI extracted at 36 weeks' gestation, and p value. Birth weight will be analysed by fitting a linear regression model. The primary neonatal hypothesis will be evaluated by estimating the absolute difference in mean birth weight between intravenous iron and standard-of-care (oral iron) along with a corresponding 95% CI and p value. Secondary repeated time point binary outcomes will be analysed similarly to anaemia, and secondary single time point continuous outcomes will be analysed similarly to birth weight. Secondary, single time point binary outcomes (eg, suboptimal pregnancy outcomes) will be analysed using a log-binomial regression model and secondary, multiple time point continuous outcomes (eg, maternal haemoglobin concentration) will be analysed using a likelihood-based longitudinal data analysis model.[27] Appropriate transformations may be applied to the variables before fitting the model if considered skewed (eg, ferritin). Additional analyses using multiple imputation will be performed to handle missing data. Results will be compared with the main analysis to investigate the findings' robustness to the missing data assumptions. Safety, including AEs, infections and clinic visits, will be presented for the mothers and neonates, respectively. The proportion of study participants with at least one safety outcome will be compared between groups using a log-binomial regression model. A Poisson model with robust standard errors will be fitted if

there is a non-convergence of the log-binomial model for efficacy or safety outcomes. Exploratory subgroup analyses (eg, by parity, site, iron deficiency) will be performed for maternal and neonatal outcomes, irrespective of their findings. The analyses models for all study outcomes will adjust for the randomisation stratification variable of the site as a main effect.

## Data management

Data collected from the subjects are recorded in digital case report forms using tablets. Relevant hard copy patient hospital files are scanned for reference and stored digitally, securely. An independent Data and Safety Monitoring Board has been established to regularly review the trial's progress and blinded and unblinded results. The Research Support Centre at the College of Medicine performs independent monitoring of the study on behalf of the sponsor. No interim analysis is planned.

## ETHICS AND DISSEMINATION

The College of Medicine Research Ethics Committee (COMREC), Blantyre, Malawi, and the Walter and Eliza Hall Institute of Medical Research Human Research Ethics Committee, Melbourne, Australia, approved this study. In addition, an evaluation was done by the Regional Ethics Committee West of Norway. They advised that the study was not subject to the Norwegian Health Research Act and that ethical review from this committee was not required. The trial is approved by the Malawian Pharmacy and Medicines Regulatory Authority (PMRA/CTRC/III/25052018100). Important protocol modifications such as changes to eligibility criteria or outcomes are reported to the ethics committees, regulatory authorities in Malawi, investigators, trial participants, trial registries. Witnessed informed written consent in English or Chichewa language is obtained from each participant before conducting any study-related procedure. The study is conducted under The International Conference on Harmonisation of Technical Requirements for Registration of Pharmaceuticals for Human Use guidelines for 'good clinical practice' and the Declaration of Helsinki. The results will be presented to and shared with the local community that hosted and enabled the research, and also to the international fora. We will publish in peer-reviewed scientific journals and report to relevant policymaking bodies such as the Malawi Ministry of Health.

## DISCUSSION

Maternal anaemia prevalence is highest in sub-Saharan Africa and South Asia, and current approaches to control it are failing. This trial will provide high quality, African-based evidence for clinicians, policymakers and donors on a role for modern intravenous iron for antenatal anaemia in low-income settings. This is the largest RCT programme assessing new intravenous iron formulations in pregnancy and is of major international significance in developing new global guidelines for anaemia in pregnancy. The results of this study will be disseminated to local and national medical authorities, policymakers, and be disseminated to the global research community, technical agencies and international government bodies via peer-reviewed journals and at international scientific fora.

**Author affiliations**
[1]Department of Public Health, School of Public Health and Family Medicine, University of Malawi, College of Medicine, Blantyre, Malawi
[2]Department of Nutrition and Infectious Diseases, Training and Research Unit of Excellence (TRUE), Blantyre, Malawi
[3]Centre for Epidemiology and Biostatistics, University of Melbourne School of Population and Global Health, Melbourne, Victoria, Australia
[4]Department of Infectious Diseases, The University of Melbourne, Melbourne, Victoria, Australia
[5]Population Health and Immunity Division, Walter and Eliza Hall Institute of Medical Research, Melbourne, Victoria, Australia
[6]Department of Global Public Health and Primary Care, University of Bergen, Bergen, Norway
[7]Department of Obstetrics and Gynaecology, University of Malawi, College of Medicine, Blantyre, Malawi
[8]Department of Medicine, Peter Doherty Institute for Infection and Immunity, University of Melbourne, Melbourne, Victoria, Australia
[9]Department of Obstetrics and Gynaecology, Zomba Central Hospital, Zomba, Malawi
[10]Malaria Alert Centre, University of Malawi, College of Medicine, Blantyre, Malawi
[11]Department of Health Promotion, Education, and Behavior, University of South Carolina, Arnold School of Public Health, Columbia, South Carolina, USA
[12]Diagnostic Haematology and Clinical Haematology, The Peter MacCallum Cancer Centre, The Royal Melbourne Hospital, Melbourne, Victoria, Australia

**Acknowledgements** We acknowledge all the study participants for their willingness to participate in this study. We appreciate the dedication of the research staff in Blantyre, Zomba and Melbourne. We are grateful for the support we continue to receive from the District Health Offices in Blantyre and Zomba districts of Malawi. We acknowledge the collaboration with the Zomba Central Hospital, especially the office of the Hospital Director. We thank Dr Alinune Kabaghe and Dr Josephine Banda, who independently packed the trial drugs and delivered them to research sites. We also acknowledge the Dr Kamiza Histopathology Laboratory for analysing the placental samples.

**Contributors** KSP and S-RP were involved in conception and trial design. MNM wrote the first draft of the paper. MNM, GM, EM, SB, RH and RA were involved in drafting of the article. SB and RH provided statistical expertise. MNM, GM, EM and KB were involved in study implementation and data acquisition. BR, JS, WS, SR, JC and LL were involved in critical revision of the article for important intellectual content. All the authors were involved in final approval of the article. Preparing study design, collection, management, analysis and interpretation of data; writing of the report; and the decision to submit the report for publication is the responsibility of the study sponsor. The study funder, the Bill and Melinda Gates Foundation, had no role in the decision to publish.

**Funding** This study is funded by the Bill & Melinda Gates Foundation (OPP1169939).

**Competing interests** None declared.

**Patient consent for publication** Not applicable.

**Provenance and peer review** Not commissioned; externally peer reviewed.

**ORCID iDs**
Martin N Mwangi http://orcid.org/0000-0002-8358-4448
Sabine Braat http://orcid.org/0000-0003-1997-3999
William Stones http://orcid.org/0000-0003-0699-2381
Jobiba Chinkhumba http://orcid.org/0000-0002-1921-619X

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
