## [Reviewer comments · BMJ Open]

ARTICLE DETAILS

TITLE (PROVISIONAL)	Protocol for a multicentre, parallel-group, open-label randomised controlled trial comparing ferric carboxymaltose with the standard of care in anaemic Malawian pregnant women – The REVAMP trial
AUTHORS	Mwangi, Martin; Mzembe, Glory; Moya, Ernest; Braat, Sabine; Harding, Rebecca; Robberstad, Bjarne; Simpson, Julie; Stones, William; Rogerson, Stephen; Biselele, Kabeya; Chinkhumba, Jobiba; Larson, Leila; Ataíde, Ricardo; Phiri, Kamija; Pasricha, Sant-Rayn

VERSION 1 – REVIEW

REVIEWER	Akshay Shah University of Oxford, Radcliffe Department of Medicine
REVIEW RETURNED	28-May-2021

GENERAL COMMENTS	Thank you for the invitation to review this excellent manuscript. It is very well written - well justified, reported, and clear targets. I have a few minor suggestions, which I hope will help improve the clarity of the manuscript: Introduction - 2nd paragraph - the authors describe the failure and limitations of oral iron. They could perhaps highlight recent advances in oral iron dosing such as once-daily/alternate day dosing which may be more efficacious (https://doi.org/10.1016/S2352-3026(17)30182-5) Methods - Inclusion Hb <10 g/dL could benefit from further justification as it slightly deviates from guideline definitions of anaemia (WHO <11 g/dL, BSH <10.5 in second trimester), although I appreciate these guideline definitions are historic and with limitations of their own- Inclusion criteria - it is unclear if multiple gestation pregnancies (e.g. twins) are included or excluded? It would make sense to include them as they are at higher-risk of developing anaemia.- Primary outcome of improvement in Hb is an efficacy outcome and not necessarily patient-centred. This could benefit from further justification - is the correction of anaemia on the causal pathway to improved maternal and fetal outcomes?- Laboratory measurements - the manuscript may benefit from some additional details on sample handling, storage and processing (e.g. in a central laboratory?)- Infection reporting - this is a concern in this setting and a definition of how infection will be diagnosed (e.g. clinical discretion, standardised definitions) will be informative. Other minor points: One recent trial on IV iron in pregnancy have been published which the authors may consider including in the discussion and how this study compares with them:
---

	- https://www.thelancet.com/journals/langlo/article/PIIS2214-109X(19)30427-9/fulltext
--	---

REVIEWER	Ghulam N Bader University of Kashmir
REVIEW RETURNED	25-Jun-2021

GENERAL COMMENTS	The manuscript "Protocol for a multicentre, parallel-group, open-label randomised controlled trial comparing ferriccarboxymaltose with the standard of care in anaemic Malawian pregnant women – The REVAMP trial claims that This trial will provide high quality, African-based evidence on a role for modern IV iron for antenatal anaemia in low-income settings, Whereas, the drug ferrichydroxy-polymaltose has undergone many trials and its safety and efficacy has been proven in pregnant women. please refer latest paper " Efficacy and safety of intravenous ferric carboxymaltose compared with oral iron for the treatment of iron deficiency anaemia in women after childbirth in Tanzania: a parallel-group, open-label, randomised controlled phase 3 trial" by Fiona Vanobberghen et al with a similar claim "Intravenous iron substitution with ferric carboxymaltose was safe and yielded a better haemoglobin response than oral iron.the Authors add that "To our knowledge, this is the first study to provide evidence of the benefits and safety of intravenous iron substitution in a low-income setting." The Lancet Volume 9, ISSUE 2, e189-e198, February 01, 2021. Another study titled "Efficacy and safety of oral iron(III) polymaltose complex versus ferrous sulfate in pregnant women with iron-deficiency anemia: A multicenter, randomized, controlled study", The journal of maternal-fetal & neonatal medicine: 24(11):1-6, has similar conclusions. so the study is not the first of its kind in African/low income settings,where the malarial infections cause frequently IDA in pregnant women. This comparative rREVAMP,multicentric trial protocol has its advantages in that, being multicentric, large number of population are included in study, which minimises the limitations. however, the manuscript needs english editing.
--

REVIEWER	Sutapa B Neogi Public Health Foundation of India
REVIEW RETURNED	19-Jul-2021

GENERAL COMMENTS	The protocol is very well written. However it requires some major changes: There is enough evidence now that mild anemia in second trimester is beneficial, the best range is Hb 9.5- 10.5 gm%. Also there is growing body of evidence that moderate anemia can be tackled using standard therapy if initiated early. Therefore I would recommend that inclusion criteria be reduced to Hb <7 gm%. Accordingly, sample size would change. Since there are conflicting reports on the safety profile of FCM, approval from Drug regulatory body is mandatory Also the source of the drug- whether it is public or private source, name of the manufacturer and country should be clearly specified in the protocol
--

VERSION 1 – AUTHOR RESPONSE

Reviewer1:

Dr. Akshay Shah, University of Oxford

Comments to the Author:

Thank you for the invitation to review this excellent manuscript. It is very well written - well justified, reported, and clear targets. I have a few minor suggestions, which I hope will help improve the clarity of the manuscript:

We thank the reviewer for their feedback.

Introduction

- 2nd paragraph - the authors describe the failure and limitations of oral iron. They could perhaps highlight recent advances in oral iron dosing such as once-daily/alternate day dosing which may be more efficacious ([https://doi.org/10.1016/S2352-3026\(17\)30182-5](https://doi.org/10.1016/S2352-3026(17)30182-5))

We thank the reviewer for this point. There have been very interesting advances in the timing and dosing of oral-iron formulations, however, we should emphasise that i) alternate day dosing has not been tested in pregnancy, and that ii) constant and reliable access to iron through health centres, as well as patient compliance are very important components of the efficacy of oral-iron as an intervention. Nevertheless, we agree and have added the following text:

“However, oral iron is often poorly tolerated due to gastrointestinal adverse effects,¹⁰ limiting adherence; lower doses with intermitted dosing have been used in non-pregnant women.¹¹”

Methods

- Inclusion Hb <10 g/dL could benefit from further justification as it slightly deviates from guideline definitions of anaemia (WHO <11 g/dL, BSH <10.5 in second trimester), although I appreciate these guideline definitions are historic and with limitations of their own.

Our primary concern was the lack of a clear clinical threshold for those women with mild anaemia (Hb>10 but <11 g/dL). We selected an Hb<10g/dL indicating moderate to severe anaemia, which we reasoned met an appropriate need for therapy. We have amended the text as follows:

“We excluded women with mild anaemia as thresholds to distinguish mild anaemia from health are indistinct, and because moderate and severe anaemia have an increased link to adverse maternal and child health outcomes.²⁵”

- Inclusion criteria - it is unclear if multiple gestation pregnancies (e.g. twins) are included or excluded? It would make sense to include them as they are at higher-risk of developing anaemia.

Under inclusion criteria (manuscript pg.11), we stipulate 'confirmed singleton pregnancy at 13-26 weeks' gestation', thus participants must not have a multiple pregnancy. Whilst we agree women with multiple pregnancies are at higher risk of anaemia, they would represent influential outliers in terms of outcomes – including birth weight, small for gestational age, gestation duration and pregnancy risk – and thus we have chosen not to include them.

- Primary outcome of improvement in Hb is an efficacy outcome and not necessarily patient-centred. This could benefit from further justification - is the correction of anaemia on the causal pathway to improved maternal and fetal outcomes?

We agree that haemoglobin is a biomarker. Our choice of haemoglobin as a primary outcome was influenced by the paucity of pre-existing data to power the study on a different outcome, given this is the initial trial in this space. However, improvement in haemoglobin may be expected to correlate with benefits in both maternal and infant outcomes. We have added the following text:

“This outcome evaluates the performance of the study intervention in helping women reach labor with optimal tissue oxygenation and resilience.”

We emphasise, though, that we have several secondary outcomes which are patient-centred and a rich array of these are listed in the manuscript, capturing critical functional outcomes including birth weight, gestation duration and perinatal adverse outcomes.

- Laboratory measurements - the manuscript may benefit from some additional details on sample handling, storage and processing (e.g. in a central laboratory?)

We agree and have added the following text (Manuscript pg. 19):

“Laboratory Procedures

Venous blood is measured for haemoglobin concentration using an automated analyser (Sysmex, XP 300 Series, Sysmex Corporation, Kobe, Japan), for which daily two-level controls are run and recorded. Serum is separated by centrifugation and stored at -80 degrees Celsius. Samples will be batched and assayed for ferritin, C-reactive protein, and phosphate in Meander Medical Centre laboratory, accreditation number M040, EN ISO 15189:2012 (Amersfoort, The Netherlands).”

- Infection reporting - this is a concern in this setting and a definition of how infection will be diagnosed (e.g. clinical discretion, standardised definitions) will be informative.

We agree and tried to clarify by adding the following text (Manuscript pg. 21):

“Infections will be reported during unplanned sick visits. Clinical malaria will be defined clinically in women who present with fever and a positive malaria test. Diarrhoea will be defined in women with more than three loose stools per day. Other clinical diagnoses will be made according to local health manuals. “

Other minor points:

One recent trial on IV iron in pregnancy have been published which the authors may consider including in the discussion and how this study compares with them:

[https://www.thelancet.com/journals/langlo/article/PIIS2214-109X\(19\)30427-9/fulltext](https://www.thelancet.com/journals/langlo/article/PIIS2214-109X(19)30427-9/fulltext)

This is well noted and we agree it warrants a mention. We have added the following text to the Introduction, where we feel this context sits best:

“Studies have evaluated the role of older forms of intravenous iron in pregnancy in low-income settings such as India,^{20 21} whilst other studies have demonstrated the feasibility of using FCM in the post-partum period in sub-Saharan Africa²² but modern formulations capable of delivering a rapid total-dose infusion have not yet been studied in women in pregnancy in low income countries.”

Reviewer 2:

Dr. Ghulam N Bader, University of Kashmir

Comments to the Author:

The manuscript "Protocol for a multicentre, parallel-group, open-label randomised controlled trial comparing ferric carboxymaltose with the standard of care in anaemic Malawian pregnant women – The REVAMP trial claims that This trial will provide high quality, African-based evidence on a role for modern IV iron for antenatal anaemia in low-income settings, Whereas, the drug ferrichydroxypolymaltose has undergone many trials and its safety and efficacy has been proven in pregnant women. please refer latest paper " Efficacy and safety of intravenous ferric carboxymaltose compared with oral iron for the treatment of iron deficiency anaemia in women after childbirth in Tanzania: a parallel-group, open-label, randomised controlled phase 3 trial" by Fiona Vanobberghen et al with a similar claim "Intravenous iron substitution with ferric carboxymaltose was safe and yielded a better haemoglobin response than oral iron.

We agree that the trial by Vanobberghen et al. evaluated the safety and efficacy of delivering FCM in a low-income setting, however, they evaluated the role of FCM in assisting women with their postnatal anaemia recovery and did not address its role in assisting women with overcoming antenatal anaemia nor did it evaluate the potential effects for mother and child. However, we have included this citation in the paper:

“Studies have evaluated the role of older forms of intravenous iron in pregnancy in low-income settings such as India,^{20 21} whilst other studies have demonstrated the feasibility of using FCM in the

post-partum period in sub-Saharan Africa²² but modern formulations capable of delivering a rapid total-dose infusion have not yet been studied in women in pregnancy in low income countries.”

(...) the Authors add that "To our knowledge, this is the first study to provide evidence of the benefits and safety of intravenous iron substitution in a low-income setting." The Lancet Volume 9, ISSUE 2, e189-e198, February 01, 2021. Another study titled "Efficacy and safety of oral iron(III) polymaltose complex versus ferrous sulfate in pregnant women with iron-deficiency anemia: A multicenter, randomized, controlled study", The journal of maternal-fetal & neonatal medicine: 24(11):1-6, has similar conclusions. so the study is not the first of its kind in African/low income settings, where the malarial infections cause frequently IDA in pregnant women.

The study by Ortiz et al. (The journal of maternal-fetal & neonatal medicine: 24(11):1-6), conducted in Argentina and Colombia, did not evaluate intravenous iron-formulations nor did it take place in areas endemic to malaria.

This comparative rREVAMP, multicentric trial protocol has its advantages in that, being multicentric, large number of population are included in study, which minimises the limitations. however, the manuscript needs english editing.

We thank the reviewer for this feedback and will carefully proofread the manuscript prior to resubmission.

Reviewer 3:

Dr. Sutapa B Neogi, Public Health Foundation of India

Comments to the Author:

The protocol is very well written. However it requires some major changes:

There is enough evidence now that mild anemia in second trimester is beneficial, the best range is Hb 9.5- 10.5 gm%. Also there is growing body of evidence that moderate anemia can be tackled using standard therapy if initiated early. Therefore I would recommend that inclusion criteria be reduced to Hb <7 gm%. Accordingly, sample size would change.

We thank the reviewer for this comment. We agree that elevated haemoglobin levels in pregnancy correlate with poorer outcomes, however, we are uncertain that mild anaemia is protective, based on recent systematic reviews for example Melissa et al 2019 (<https://nyaspubs.onlinelibrary.wiley.com/doi/10.1111/nyas.14093>) which demonstrates an increased risk of adverse pregnancy outcomes in even mild to moderately anaemic women. Importantly, our trial has already been approved by ethics, by the drug regulator, and has already commenced, thus we can no longer adjust the recruitment criteria in the protocol. We will certainly consider the reviewer's point when planning any future studies.

Since there are conflicting reports on the safety profile of FCM, approval from Drug regulatory body is mandatory

We agree, and the trial is approved by the regulator. We have added the following text on Pg. 24:

“The trial is approved by the Malawian Pharmacy and Medicines Regulatory Authority (PMRA/CTRC/III/25052018100).”

Also the source of the drug- whether it is public or private source, name of the manufacturer and country should be clearly specified in the protocol

We have provided the following text on pg. 12 under Trial Interventions:

“Ferric carboxymaltose, manufactured by Vifor Pharma, was purchased at full-price from Aspen Pharma in Australia and shipped to Malawi.”